# Amphibian chytridiomycosis outbreak dynamics are linked with host skin bacterial community structure

Kieran A. Bates [1,2], Frances C. Clare[2], Simon O'Hanlon[1], Jaime Bosch[3], Lola Brookes[2], Kevin Hopkins[2], Emilia J. McLaughlin[2], Olivia Daniel[1], Trenton W. J. Garner [2], Matthew C. Fisher [1] & Xavier A. Harrison[2]

Host-associated microbes are vital for combatting infections and maintaining health. In amphibians, certain skin-associated bacteria inhibit the fungal pathogen *Batrachochytrium dendrobatidis* (*Bd*), yet our understanding of host microbial ecology and its role in disease outbreaks is limited. We sampled skin-associated bacteria and *Bd* from Pyrenean midwife toad populations exhibiting enzootic or epizootic disease dynamics. We demonstrate that bacterial communities differ between life stages with few shared taxa, indicative of restructuring at metamorphosis. We detected a significant effect of infection history on metamorph skin microbiota, with reduced bacterial diversity in epizootic populations and differences in community structure and predicted function. Genome sequencing of *Bd* isolates supports a single introduction to the Pyrenees and reveals no association between pathogen genetics and epidemiological trends. Our findings provide an ecologically relevant insight into the microbial ecology of amphibian skin and highlight the relative importance of host microbiota and pathogen genetics in predicting disease outcome.

[1] Department of Infectious Disease Epidemiology, Imperial College London, London W2 1PG, UK. [2] Institute of Zoology, Zoological Society of London, Regent's Park, London NW1 4RY, UK. [3] Museo Nacional de Ciencias Naturales, CSIC, Jose Gutierrez Abascal 2, 28006 Madrid, Spain. Correspondence and requests for materials should be addressed to K.A.B. (email: k.bates14@imperial.ac.uk) or to X.A.H. (email: xavier.harrison@ioz.ac.uk)

The communities of bacteria resident on multicellular organisms are of significant importance to host health[1]. In addition to their involvement in essential host physiological processes[2–5], bacterial interactions with invading pathogens are key in defining disease outcome[6]. Commensal bacteria may confer protection against pathogens through limiting pathogen adhesion to host cells[7], resource competition[8], interacting with the host immune system[9], or production of antimicrobial compounds[10]. Conversely, some commensal bacteria interact synergistically with pathogens exacerbating infection[11]. At a community level, disruption of the microbiome (a process termed dysbiosis) can negatively impact host health bringing about pathophysiological changes and in some instances facilitating infection by opportunistic pathogens[12].

The emerging infectious disease, chytridiomycosis caused by the fungal pathogen Batrachochytrium dendrobatidis (Bd) is known to affect over 500 amphibian species worldwide[13]. Bd infects keratinised skin of post-metamorphic anurans where it disrupts osmoregulation leading to cardiac abnormalities and in some cases death[14]. Larval anurans exhibit little Bd-induced pathology since the skin is deficient of keratin and infection is restricted to the mouthparts[15]. Prior studies have gained insight into the drivers of Bd infection dynamics by examining host and pathogen biology in addition to environmental effects on disease outcome[16–18]. Recently attention has shifted to include the role of skin-associated bacteria in infection based on findings that certain taxa inhibit Bd[19,20]. Other studies have shown associations between bacterial communities and infection in the field[21–24]; however, research has largely focused on within-population infection dynamics and on species solely from the Americas. With the exception of an amphibian study system in the Western United States[21,24], most studies to date are based on short-term surveillance programmes and therefore lack vital information such as time of initial Bd emergence. In addition, while previous studies have demonstrated that pathogen genetics can underpin Bd virulence[17] and disease dynamics in the field[25], the relative importance of the microbiome and pathogen genetics have not been simultaneously investigated in the wild. Therefore, it remains to be established whether pathogen genetics and microbiome covary and the extent to which they are associated with disease dynamics. Consequently prior studies have been limited in their ability to examine the relationships between host bacterial communities and disease dynamics in complex ecosystems.

In this study, we use 16s metabarcoding to disentangle the relationship between skin bacterial community structure, life stage and Bd infection in wild common midwife toads (Alytes obstetricans). We also use whole genome sequencing to investigate the spatial and temporal genetic structure of Bd isolates from host populations exhibiting distinct disease dynamics. Our work focuses on a long-term study system in the Pyrenees of France and Spain in which infection has been monitored continuously for over 10 years[26,27]. While A. obstetricans is native throughout Europe, it has shown catastrophic declines in the Pyrenees with

several populations undergoing mass mortality events[26–28]. In recent years however, certain populations have shown signs of tolerance to infection and an associated resurgence in abundance consistent with the establishment of enzootic disease dynamics. Conversely, other populations infected at approximately the same time remain in decline and show high susceptibility to Bd indicative of epizootic dynamics.

Our results demonstrate that Bd in the Pyrenees arose from a single introduction and isolates from enzootic and epizootic populations are not genetically distinct. Significant differences were however found in the skin microbiome of amphibians, with life stage, population and Bd infection dynamics exhibiting a strong association with bacterial community structure. We further show that the predicted function of metamorph skin bacterial communities differs based on disease dynamics. We identify a conserved set of bacterial taxa associated with different disease states that may represent a dysbiosis contributing to epizootic dynamics, or selection of protective symbionts leading to an enzootic disease state. Our findings show that among populations with little spatial separation, bacterial community structure but not pathogen genetics is associated with divergent disease outcomes.

## Results

**Host skin microbiota differs by life stage and population**. We collected midwife toad larvae and metamorphs in August 2015 from five populations in the Pyrenees mountains of France and Spain (Table 1). Each population has been part of a long-term study in which population abundance and infection have been monitored (Supplementary Table 1, Supplementary Fig. 1).

We quantified the bacterial communities of amphibian skin and environmental samples by sequencing the V4 region of the 16s rRNA gene. To determine the importance of life stage and the environment in shaping the amphibian microbiome, we first calculated the proportion of bacterial operational taxonomic units (OTUs) common to different samples. We detected a total of 11,112 OTUs, of which 17.3% were shared by larvae and metamorphs, 10.8% were shared by larvae and the environment, 10.1% were shared by metamorphs and the environment, and 7.8% were common to all sample types (Fig. 1a). Similar results were obtained when analysing populations on an individual basis (Supplementary Table 2). In larvae and metamorphs, over 98% of OTUs had a relative abundance of less than 1% in the environment.

To assess differences in bacterial community structure between amphibian life stages and the environment, we calculated alpha and beta diversity metrics. Shannon diversity differed only among samples from Acherito (ANOVA $F_{(3,30)} = 6.462$, $P = 0.002$) in which sediment had a significantly lower value relative to larvae, metamorphs and lake water (Tukey's test: sediment-larva $P = 0.001$; sediment-metamorph $P = 0.002$; sediment-water $P = 0.04$, Fig.1b, Supplementary Fig. 2). Both life stage and population were associated with variation in beta diversity of

---

**Table 1 Sample summary of each lake**

| Population | Larvae | Metamorphs | Environmental samples | Elevation (m) | Bd first detected | Bd infection status |
|---|---|---|---|---|---|---|
| Acherito | 16 | 14 | Water sediment | 1869 | 2004 | Enzootic |
| Lhurs | 15 | 11 | Water sediment | 1691 | 2009 | Enzootic |
| Puits d'Arious | 17 | 7 | Water sediment | 1867 | 2006 | Enzootic |
| Ansabere | 15 | 3 | Water sediment | 1859 | 2005 | Epizootic |
| Arlet | 0 | 16 | Water sediment | 1986 | 2005 | Epizootic |

N = 2 environmental samples each for water and sediment at each population

---

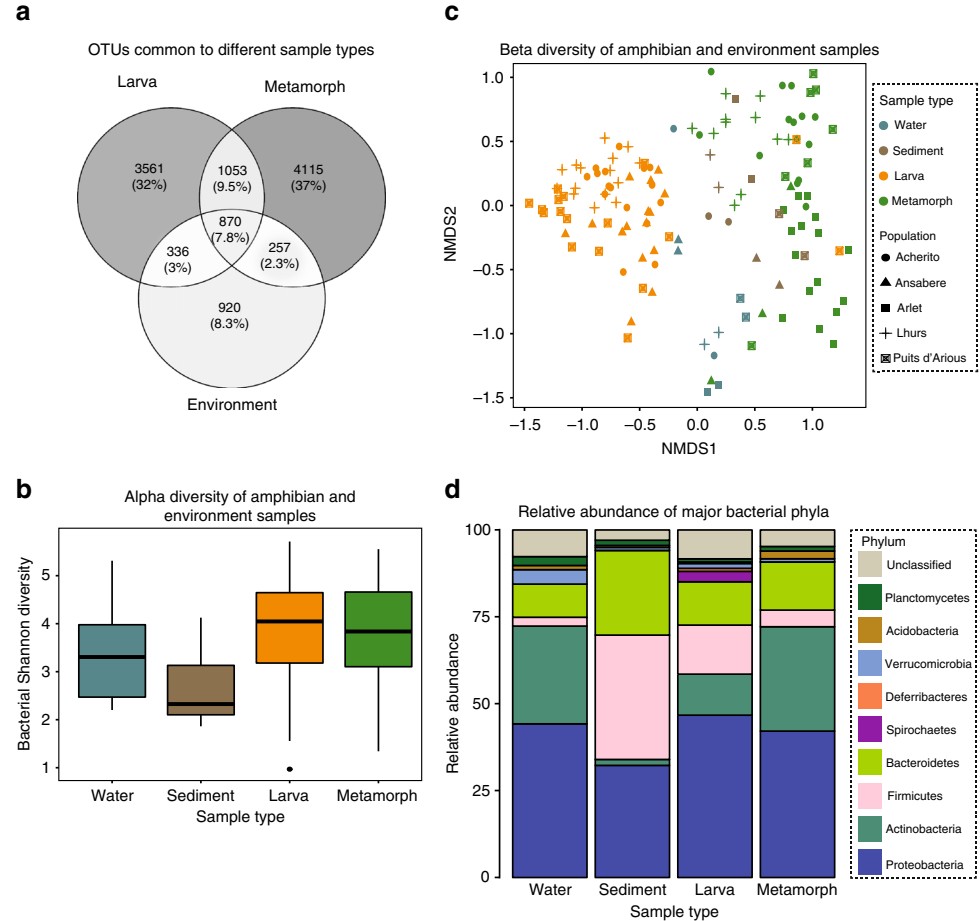

**Fig. 1** Bacterial communities of *Alytes obstetricans* skin across life stages and in environmental samples. **a** OTUs common among life stages and environmental samples. **b** Shannon diversity by sample type (boxes represent 25 and 75 percentile, the horizontal line is the median, whiskers are maximum and minimum values). **c** NMDS plot of bacterial communities based on life stage, environmental samples and population. **d** Stacked bar chart of the ten most abundant phyla for lake water, sediment, larvae and metamorphs. Sample sizes were: $n = 63$ (larvae); $n = 51$ (metamorphs); $n = 20$ (environment)

amphibian skin (Life Stage: PERMANOVA Pseudo-$F_{(1,112)} = 19.821$, $P = 0.001$; Population: PERMANOVA Pseudo-$F_{(4,109)} = 6.6086$, $P = 0.001$; Fig. 1c). There was a significant difference in beta diversity of environmental samples compared to larvae and metamorphs (Larva-Environment: PERMANOVA $F_{(1,81)} = 12.372$, $P = 0.001$; Metamorph-Environment: PERMANOVA $F_{(1,69)} = 6.0735$, $P = 0.001$; Supplementary Table 3).

The composition of bacterial communities also differed between amphibian life stages and the environment. Larvae had a higher relative abundance of Firmicutes and Spirochaetes compared to metamorphs, while metamorphs were enriched for Actinobacteria and Acidobacteria. Of the environmental samples, water had a higher relative abundance of Verrucomicrobia compared to sediment and amphibian skin, while sediment exhibited the highest relative abundance of Firmicutes and Bacteroidetes of any sample type (Fig. 1d, Supplementary Fig. 3).

**Microbiota differ by metamorph population and disease state.** The infection history of each Pyrenean population is well documented with estimates of *Bd* emergence for each locality. Populations have shown different responses to infection over time with Arlet and Ansabere exhibiting epizootic disease dynamics based on continued declining larval population abundance coinciding with *Bd* infection. Meanwhile, though Lhurs, Acherito and Puits d'Arious displayed *Bd*-associated decline in the initial years from

when *Bd* was first detected, all have recently shown the hallmarks of enzootic infection characterised by stable larval population abundance in the presence of sustained *Bd* infection (Fig. 2, Supplementary Table 1, Supplementary Fig. 1). Analysis of the relationship between *Bd* infection dynamics and bacterial community structure was performed for metamorphs only since we were not able to collect larvae from one epizootic population (Arlet). *Bd*-associated mortality occurs during or after metamorphosis; so we consider the bacterial communities of metamorph skin to be most informative in terms of response to infection.

Enzootic populations exhibited lower *Bd* prevalence and significantly reduced infection intensity compared to epizootic populations (Chi-sq = 59.855, df = 4, $P < 0.0001$, Fig. 2, Supplementary Table 4). Bacterial Shannon diversity was reduced in epizootic populations compared to enzootic populations (ANOVA $F_{(4,46)} = 6.837$, $P = 0.0002$; Tukey's tests: Ansabere–Acherito, $P = 0.008$; Ansabere–Lhurs, $P = 0.05$; Ansabere–Puits d'Arious $P = 0.06$; Arlet–Acherito, $P = 0.001$; Arlet–Lhurs, $P = 0.04$; Arlet–Puits d'Arious, $P = 0.08$; Fig. 3a). Beta diversity of host skin bacterial communities based on Bray–Curtis distances differed according to both population (PERMANOVA Pseudo-$F_{(4,46)} = 3.175$, $P = 0.001$) and infection dynamic (PERMANOVA Pseudo-$F_{(1,49)} = 4.5125$, $P = 0.001$, Fig. 3b).

Bacterial community composition differed in enzootic and epizootic populations (Fig. 3c). Enzootic populations had a higher

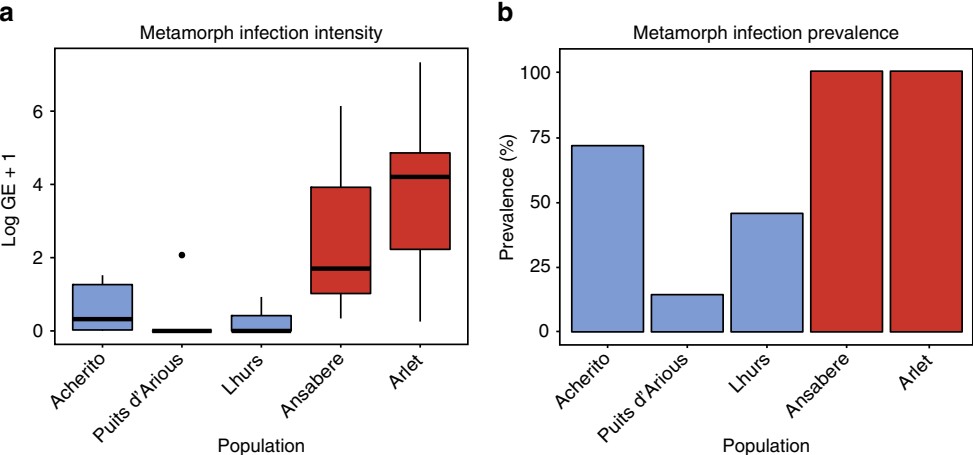

**Fig. 2** *Bd* infection profiles of *Alytes obstetricans* metamorph populations. **a** Boxplot of *Bd* infection intensity in metamorphs sampled in 2015 showing reduced infection intensity in enzootic populations (Chi-sq = 59.855, P < 0.0001). Boxes represent 25 and 75 percentile, the horizontal line is the median and whiskers are maximum and minimum values of infection intensity. **b** Bar chart of *Bd* infection prevalence in metamorphs. Enzootic populations are coloured in blue, epizootic populations are coloured in red. Sample sizes were: Enzootic = 32 (Acherito = 14, Lhurs = 11, Puits d'Arious = 7); Epizootic = 19 (Arlet = 16, Ansabere = 3)

relative abundance of OTUs from the phyla Acidobacteria, Verrucomicrobia and Planctomycetes, while epizootic populations had higher relative abundance of Bacteroidetes. Arlet, which has shown the highest mortality rate of any lake, featured an increased relative abundance of Chlamydiae, a phylum whose members include obligate intracellular pathogens[29]. The differences in bacterial community structure and composition between enzootic and epizootic populations are not an artefact of spatial non-independence given the lack of clustering of sites displaying similar disease dynamics (Fig. 3d)

We performed indicator analysis to identify bacterial taxa driving the differences in beta diversity based on both disease dynamic and population. We identified 61 indicator OTUs that were associated with enzootic dynamics and 25 OTUs associated with epizootic dynamics (Fig. 4, Supplementary Data 1). Indicator OTUs from enzootic sites spanned nine phyla and 16 classes with the orders Actinomycetales and Rhizobiales accounting for the majority of taxa. Conversely OTUs from epizootic populations represented a less diverse array of lineages with six phyla and nine classes of which Pseudomonadales and Flavobacteriales were the most abundant orders.

A range of OTUs contributed to differences in bacterial community structure between individual populations (Supplementary Data 2). Lhurs, Acherito and Puits d'Arious were characterised by 73, 51 and 29 OTUs respectively, while Arlet and Ansabere had a total of 11 and four indicator OTUs.

**Predicted function differs by population and disease state**. The predicted function of bacterial communities supported the patterns shown by bacterial taxonomic composition with significant differences based on both disease dynamic (PERMANOVA Pseudo-$F_{(1,49)}$ = 7.1993, P = 0.001) and population (PERMANOVA Pseudo-$F_{(4,46)}$ = 2.9201, P = 0.005, Supplementary Fig. 4). In addition, 11 functional features were found to be significantly differentially abundant between enzootic and epizootic populations (LEfSe LDA > 2). Nine functional features were more abundant in enzootic populations and two were more abundant in epizootic populations (Supplementary Table 5). Differential functional features in both enzootic and epizootic populations were largely associated with mineral and ion transport (Supplementary Table 5).

***Bd* genomic analysis**. Whole genome sequencing was performed on 81 *Bd* isolates collected between 2004 and 2015 (Supplementary Data 3). Phylogenetic analysis with the inclusion of *Bd* isolates collected from other European localities gave 100% bootstrap support for a single introduction of *Bd* into the Pyrenees (Fig. 5a). A principle component analysis (PCA) of identified single nucleotide polymorphisms (SNPs) revealed little temporal or spatial genetic structure among populations in the pathogen (Fig. 5b, c, Supplementary Fig. 5)

## Discussion

Our understanding of the microbial ecology of amphibian skin is rapidly increasing[21–23,30,31]. However, examining the dynamics of bacterial communities across wild populations exhibiting enzootic and epizootic disease states remains an important and understudied area[21,24]. In addition, our knowledge of how bacterial communities of amphibian skin differ between developmental stages is based on a limited number of studies[22,31,32]. In this study we address these gaps in our knowledge by characterising the skin-associated bacterial communities of amphibians from different life stages and metamorph populations displaying enzootic or epizootic disease dynamics. In addition, for the first time we investigate the relationship between *Bd* genetics and the microbiome in disease outcome by sequencing *Bd* isolates from each population. We demonstrate that host population, infection dynamics and life stage are all significant factors in shaping amphibian skin bacterial communities, with population accounting for the greatest amount of variation. While there was a clear microbiome signature corresponding to disease outcome, an association between disease state and pathogen genetics was not evident.

The bacterial community structure of amphibian skin differed significantly based on both life stage and population, accounting for 15 and 20% of variation respectively. Despite the strong effect of population, few OTUs were shared between amphibian skin and the environment, and those that were had a low relative abundance in the environment. The small proportion of OTUs shared between the host and its surrounding environment, coupled with the low abundance of environmental OTUs has been found in similar studies[30] and may be indicative of selection occurring on the skin for rare environmental bacterial species[33].

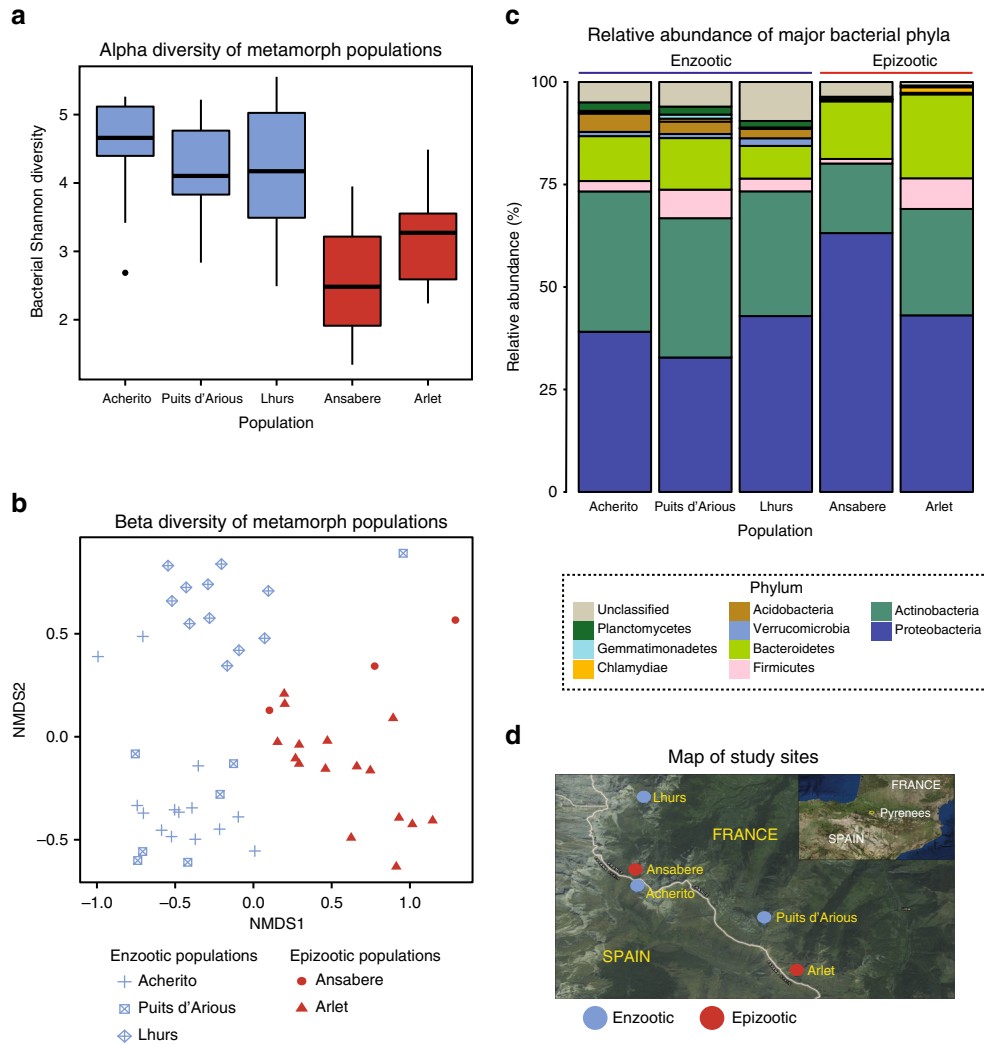

**Fig. 3** Bacterial community profiles of *Alytes obstetricans* metamorph skin from enzootic and epizootic populations. **a** Shannon diversity of metamorph populations. Epizootic populations are in red and enzootic populations are blue. Boxes represent 25 and 75 percentile; the horizontal line is the median and whiskers are maximum and minimum values. **b** NMDS plot based on Bray−Curtis distances displaying beta diversity of enzootic and epizootic populations. **c** Stacked bar chart of abundant bacterial phyla for each metamorph population. **d** Map showing location of enzootic and epizootic populations. Map generated using ArcGIS version 10.0 (http://www.esri.com/software/arcgis) with the World Imagery Basemap. Source: Esri, DigitalGlobe, GeoEye, Earthstar, Geographics, CNES/Airbus DS, USDA, USGS, AeroGRID, IGN, the GIS User Community. Sample sizes were: Enzootic = 32 (Acherito = 14, Lhurs = 11, Puits d'Arious = 7); Epizootic = 19 (Arlet = 16, Ansabere = 3)

This finding is surprising given the strong effect that host population was found to have on bacterial community structure, and existing evidence from other studies suggesting environmental reservoirs are important in shaping the vertebrate microbiome[20,34,35]. Our results may therefore reflect that host OTUs also present in the environment may be at sub-detectable levels.

Larvae and metamorphs were dominated by the phylum Proteobacteria of which an OTU belonging to the family Comamonadaceae was most abundant. Comamonadaceae have also been identified as the most common taxonomic group in other studies of North American amphibian species[22,36], suggesting some members of the core skin microbiota may be conserved across continents as has previously been shown for constituents of the tadpole gut microbiome[37]. Unlike prior studies[22,31], we found no significant difference in alpha diversity of metamorph and larval amphibians. There was however clear evidence for restructuring of skin-associated bacterial communities during metamorphosis. Specifically, there was a significant difference in beta diversity of

metamorphs and larvae and only a small proportion (17.3%) of OTUs were shared between life stages. This restructuring of the skin bacterial community is probably due to a combination of factors including the shift from a wholly aquatic to terrestrial environment and the extensive physiological changes that the skin undergoes during metamorphosis. Of particular importance is likely to be the heavily expanded repertoire of skin secretions produced post-metamorphosis that transforms the skin into a very different niche for bacteria[38].

Our findings show that while there was no distinction in pathogen genetic structure across enzootic and epizootic populations, there were clear differences in community structure and predicted function of bacteria on metamorph skin based on population and disease dynamic. Importantly, differences in bacterial community structure also reflect variation in bacterial predicted function. This finding is not indicative of functional redundancy among disparate bacterial communities and supports an active role of bacteria in host defence. Alpha diversity was significantly lower in epizootic populations compared to those

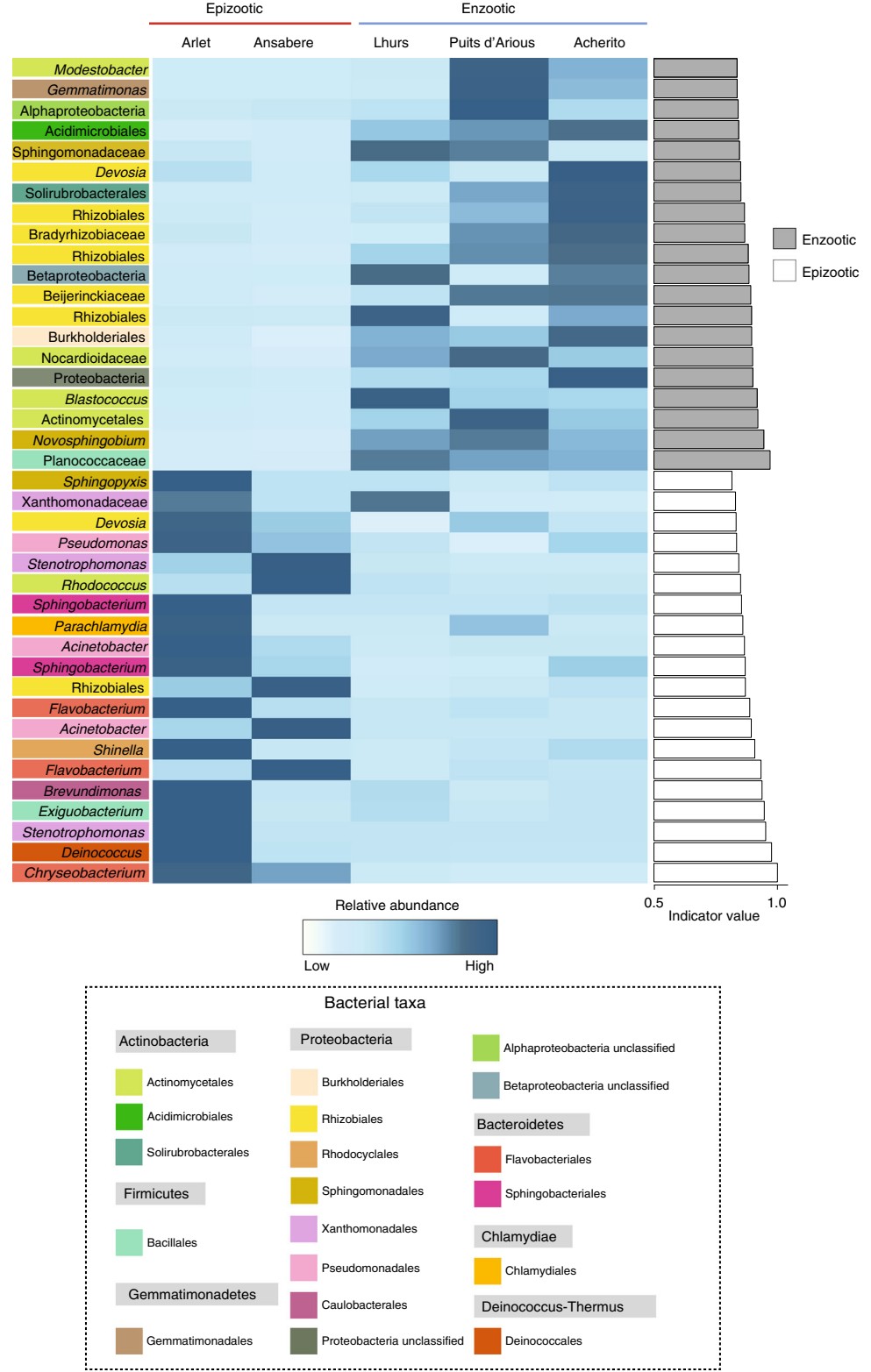

**Fig. 4** Heatmap displaying the normalised relative abundance of the top 20 bacterial OTUs from enzootic and epizootic populations identified from indicator analysis. Sample sizes were: Enzootic = 32 (Acherito = 14, Lhurs = 11, Puits d'Arious = 7); Epizootic = 19 (Arlet = 16, Ansabere = 3)

presenting enzootic dynamics. This finding corroborates that of other studies[39] and may indicate that high alpha diversity offers enhanced protection, for example through production of a larger arsenal of antifungal metabolites or by competing for resources more effectively than the invading *Bd*. High *Bd* infection intensity in epizootic populations also results in increased skin sloughing which has been shown to decrease resident bacterial community diversity[40]. Further, epizootic populations may experience a more severe inflammatory response to *Bd* and impaired immune function[41] that promotes dysbiosis. This in turn can lead to enrichment of bacteria that exploit weakened host defences and consequently outcompete beneficial symbionts leading to

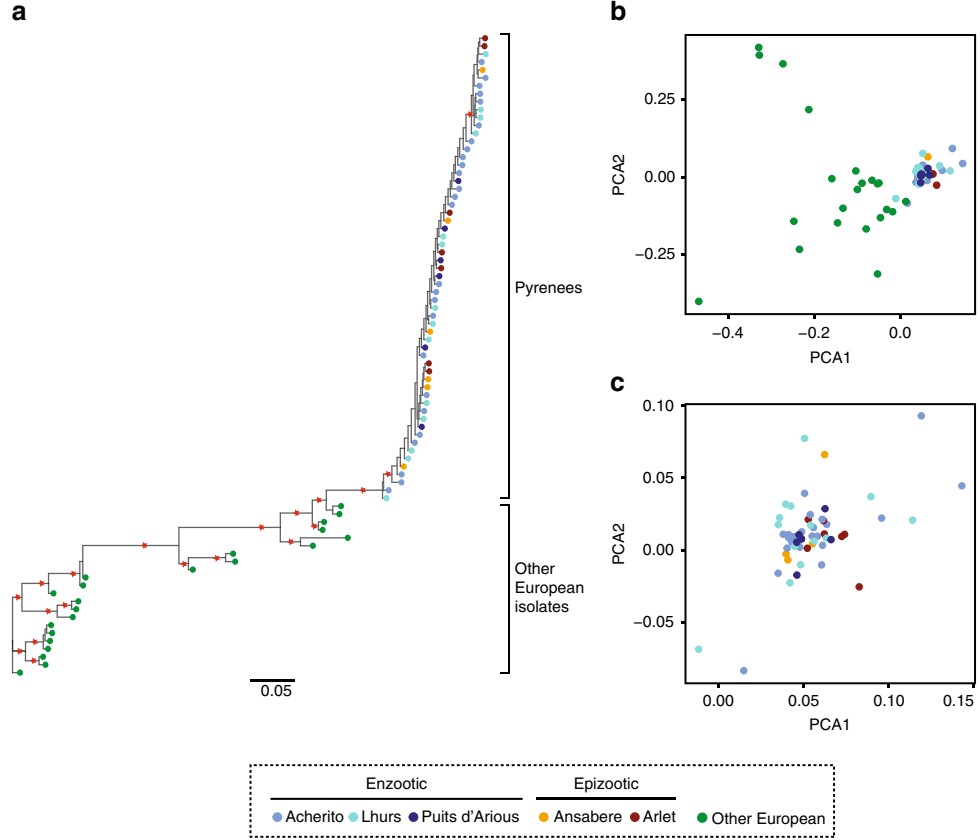

**Fig. 5** *Bd* genetics across space and time in the Pyrenees. **a** Phylogeny of *Bd* isolates collected from the Pyrenees and other European localities. Red stars indicate bootstrap values of 100. **b** PCA plot of identified SNPs from *Bd* isolates from all localities. **c** PCA plot of identified *Bd* SNPs from Pyrenean lakes only. Sample sizes were: $n = 6$ (Ansabere); $n = 7$ (Arlet); $n = 26$ (Acherito); $n = 14$ (Lhurs); $n = 6$ (Puits d'Arious), $n = 22$ (other European isolates)

dominance of a small number of opportunistic pathogenic bacterial species[42].

Indicator analysis based on population identified key taxa driving differences in beta diversity and revealed OTUs of potential importance to population-specific disease dynamics. Of particular interest is the indicator OTU profile of Lhurs, where metamorphs carried a high abundance of an OTU belonging to the *Lysobacter* genus (Supplementary Fig. 6). *Lysobacter* is inhibitory to several fungal pathogens including *Bd*[43] and anti-*Bd* functional traits have been identified for this genus. Genomic studies reveal that *Lysobacter* possesses strong chitinase functionality[44] that may make it efficient at breaking down the chitin rich cell wall of *Bd* zoosporangia. Further, one study showed that *Lysobacter gummosus* produced the metabolite 2,4-diacetylphloroglucinol (2,4-DAPG) which is strongly inhibitory to *Bd*[10]. Since Lhurs was the most recently infected population, it is unlikely to have fixed host-associated genes for resistance faster than previously infected populations that remain susceptible. Therefore, one possibility is that tolerance to *Bd* may have arisen through selection of beneficial bacteria such as *Lysobacter*. This biotic shift has the potential to occur over a relatively short temporal scale given the rapid generation time of bacteria, which can subsequently become established members of the skin bacterial community and confer protective benefits to the host.

Arlet also revealed an interesting indicator OTU profile with high abundance of an OTU belonging to the *Parachlamydia* genus (Supplementary Fig. 7a). The Chlamydiae comprise obligate intracellular pathogens that infect mucosal epithelial cells[29]. Prior studies show that co-infection with Chlamydiae and *Bd* can result in mortality and epizootic dynamics[45]. Arlet experienced a

severe population decline coinciding with *Bd* emergence[18,27] and despite being infected longer than two enzootic populations (Lhurs and Puits d'Arious) has shown no signs of recovery. Analysis of Chlamydiae across all populations shows a higher relative abundance in metamorphs than either larvae or environmental samples indicating either colonisation or proliferation at metamorphosis (Supplementary Fig. 7b). If Chlamydiae are found to induce pathology in Arlet metamorphs, then identifying whether this phylum is potentially associated with secondary infection or acting synergistically to promote *Bd* growth will be of paramount importance.

Indicator analysis based on infection dynamic identified a wide range of bacterial groups. The OTU with the highest indicator value for enzootic populations belonged to the family Planococcaceae of which several taxa are known to possess strong inhibitory capabilities against multiple pathogenic fungi including *Bd*[43,46]. Members of the Actinomycetales and Rhizobiales represented the most abundant indicator OTUs in enzootic populations. These orders are common constituents of soil[47] and have previously been documented for their anti-*Bd* activity in vitro[43]. Actinomycetales in particular are likely to be important candidates contributing to *Bd* resistance with prior studies identifying associations between members of this order and *Bd* tolerance in the wild[21,23]. The Actinomycetales are also functionally important producing an estimated 53% of bioactive microbial metabolites of known antibiotic capability[48].

Among epizootic populations, members of the Moraxellaceae, and Flavobacteriaceae were highly represented indicator OTUs that have all been associated with high *Bd* loads in prior studies[21,23,49]. All indicator OTUs assigned to the family

Moraxellaceae belonged to the *Acinetobacter* genus. Isolates of *Acinetobacter* taken from multiple amphibian species (including *A. obstetricans*) have shown a great deal of functional diversity, including both *Bd* enhancement and inhibition[43]. These results extend to in vivo studies of which one field study found *Acinetobacter* to be enriched in non-susceptible populations[23], while a controlled experiment showed members of the Moraxellaceae to be strongly associated with *Bd*-induced mortality[49]. Flavobacteriaceae are also often positively correlated with *Bd* load in wild amphibian populations[21] and in some cases enhance *Bd* growth in co-culture[43].

Overall, our findings reveal that while individual populations may be characterised by unique OTUs that likely reflect the ecology of each habitat, there exists a subset of bacterial taxa that are conserved across populations depending on disease state. This may indicate the existence of common selection pressures linked with disease dynamics across sites that are associated with enhanced success of certain taxa. Such selection pressures are likely to be complex and mediated by the host, microbe–microbe interactions and the environment[50].

One possibility is that the pathogen itself may contribute to shaping the skin bacterial community of amphibians from each population. While *Bd* from epizootic and enzootic populations lacked marked differences in genetic structure, it remains possible that there is among-population heterogeneity in the way that bacterial taxa could respond to *Bd* genotypes[51]. Differences in bacterial communities among populations exhibiting contrasting infection dynamics may also be mediated by host factors such as the composition and volume of skin secretions that may be linked to proliferation of certain bacteria[52]. If co-evolution of the host with protective symbionts has occurred, this may not only lead to tolerance of infection through direct inhibition of *Bd* but may also culminate in reduced pathogen virulence[53]. Such a scenario may provide an elegant explanation for the dynamics of the Pyrenees system whereby *Bd* persists in enzootic populations without causing pathology to the host. Ultimately, identifying the selection pressures acting on amphibian-associated bacteria, whether mediated by the host, pathogen or environment will be insightful in furthering our knowledge of microbial ecology of disease dynamics in nature. While a great body of work has focussed on in vitro studies identifying single bacterial isolates that are inhibitory to *Bd*, our findings highlight that differences in epizootic and enzootic populations are likely driven by communities of bacteria. This may be particularly important since many constituents of a bacterial community may not directly inhibit or promote *Bd* growth but instead may produce metabolic by-products that interact with *Bd*[54]. In addition, our current knowledge of bacterial phenotypic plasticity, which can be affected by factors such as community structure and other biotic and abiotic conditions[55] reinforces the necessity to shift attention from single bacterial studies to those that take into account entire communities.

Amphibians globally are facing an extinction crisis and emerging infectious diseases are a major contributor to their decline. Understanding the role of host-associated bacteria in infection will greatly enhance our knowledge of the drivers underpinning heterogeneity in disease dynamics across space and time. By demonstrating significant associations between host skin bacterial communities, life stage and infection dynamics, this study develops an insight into the microbial ecology of amphibian skin in its natural setting. In addition, the lack of association between pathogen genetics and disease outcome suggests that the microbiome may play a more important role in infection than previously expected. In this Pyrenean montane system, we identify a conserved assemblage of bacterial taxa associated with disease state, many of which have previously been identified for their

positive or negative impact on *Bd*. Despite the convincing patterns in bacterial community structure and predicted function based on disease state, our findings lack direct functional evidence to support the hypothesis that putatively beneficial or detrimental bacteria are associated with clinically relevant effects on disease outcome. In addition, other potentially important microbial groups such as Fungi were not examined in this study. Future studies should therefore seek to survey a broader range of microbial kingdoms. There is a need to move beyond functional prediction based on 16S marker genes and instead capture direct functional information, for example through metabolomics to uncover associations between microbial community structure and function in disease ecology.

## Methods

**Ethics statement.** Research was performed under licence from the Parc National des Pyrenees and the Instituto Aragonés de Gestión Ambiental.

**Sample collection.** Recently metamorphosed and larval *A. obstetricans* were captured and rinsed with sterile water to remove transient bacteria on the skin[56]. Using single sterile swabs, we sampled the skin microbiome by swabbing the body of tadpoles and metamorphs ten times, additionally swabbing the fore- and hindlimbs of metamorphs five times. To sample for *Bd* in metamorphs, we swabbed the pelvic patch and ventral surface of the hind limbs ten times. A fresh pair of sterile latex gloves was worn for each animal to avoid cross contamination of *Bd* or bacteria. Environmental water samples were collected following previously published methods[31]. For water samples a sterile swab was moved through lake water for 30 s at a depth of approximately 30 cm. Sediment samples were collected at the same location as water samples by embedding the swab into sediment for 30 s. All swabs were immediately stored on ice before being transferred to a freezer. MW100 rayon tipped dry swabs (MWE Medical Wire, Corsham, UK) were used for *Bd* and bacterial sampling. Characterisation of disease dynamics for each *A. obstetricans* population was done through long-term ecological monitoring (see Supplementary Note 1 for further details).

**DNA extraction and sample processing.** Genomic DNA was extracted from microbiome swabs using the DNeasy Blood and Tissue kit (Qiagen, Venlo, the Netherlands) according to the manufacturer's instructions. A pre-treatment with mutanolysin was included to enhance recovery of bacterial DNA[57]. DNA extracted from swabs was used to amplify the V4 region of the 16S rRNA gene using custom barcoded primers and PCR conditions adapted from a prior study[58]. PCR conditions consisted of a denaturing step of 95 °C for 15 min, followed by 28 cycles of 95 °C for 20 s, 50 °C for 60 s, 72 °C for 60 s and a final extension step of 72 °C for 10 min. Each PCR including a negative water control was performed in triplicate. Amplicons were visualised on a 2% agarose gel and pooled yielding a final per sample volume of 24 μl. Pooled amplicon DNA was purified using an Ampure XP PCR purification kit (Beckman Coulter, California, USA). Following purification, 1 μl of each combined sample was pooled into a preliminary library and the concentration was determined using Qubit fluorometric quantification (Life Technologies, California, USA). Amplicon quality and incidence of primer dimer was assessed using an Agilent 2200 TapeStation system (Agilent Technologies, California, USA). A titration run of 300 sequencing cycles was performed on an MiSeq instrument (Illumina, California, USA) to quantify the number of reads yielded per sample from the preliminary library. An equimolar concentration of each sample was then pooled into a final composite library based on the index representation from the titration run and subsequently sequenced on a 500 cycle MiSeq run with a 250 bp paired-end strategy. Genomic DNA from *Bd* swabs was extracted using a bead-beating protocol[59]. DNA extractions were diluted 1/10 before undergoing qPCR amplification with each sample run in duplicate[59] and with *Bd* standards of 100, 10, 1 and 0.1 zoospore genomic equivalents (GE). Samples with greater than 0.1 GE were considered positive for *Bd*.

**Sequence analysis.** Sequences were processed using MOTHUR[60] following a previously described method[58]. Paired-end reads were split by sample and assembled into contigs. Sequences were quality filtered by removing ambiguous base calls, removing homopolymer regions longer than 8 bp, and trimming reads longer than 275 bp. Duplicate sequences were merged and aligned with 16S reference sequences from the SILVA small-subunit rRNA sequence database[61]. A pre-clustering step grouped sequences differing by a maximum of 2 bp. Chimeric sequences were removed using UCHIME[62] as implemented in MOTHUR. 16S rRNA gene sequences were clustered into groups according to their taxonomy at the level of Order and assigned OTUs at a 3% dissimilarity level. Sequences were taxonomically classified with an 80% bootstrap confidence threshold using a naïve Bayesian classifier with a training set (version 9) made available through the Ribosomal Database Project (http://rdp.cme.msu.edu)[63]. Sequences derived from chloroplasts, mitochondria, archaea, eukaryotes or unknown reads were

eliminated. The number of sequences per sample ranged from 3345 to 252,910. To mitigate the effects of uneven sampling[64], all samples were rarefied to 3345 sequences corresponding to the size of the lowest read sample. Rarefying at a higher threshold of 10,000 sequences led to some samples being discarded but had minimal impact on results for alpha and beta diversity. Downstream analysis of OTUs was carried out using the package Phyloseq[65] in R[66] version 3.4.1.

**Infection analysis**. All statistical analyses were carried out using R[66] version 3.4.1. Analysis of infection intensity followed previously published methods[18]. Genomic equivalents were rounded to whole numbers and treated as count data. A negative binomial regression model (function 'glm.nb', package MASS[67]) was used to establish if infection intensity differed between populations. A likelihood ratio test was used to assess significance of population in the model. Tukey post-hoc testing (function 'glht', package multcomp[68]) was performed to identify populations that had significantly different infection intensities. Genomic equivalents data were visualised using boxplots and log+1 transformed for presentation purposes. Prevalence data was displayed using barplots.

**Bacterial communities across space and life history stage**. To determine the relationship between life stage, environmental bacteria and host bacterial communities, we calculated the proportion of shared and unique OTUs between host and environmental samples and between developmental stages. Water and sediment samples were combined into a single factor called 'Environment' for computing shared OTUs since we were interested in how the environment regardless of specific sample type shaped the host-associated bacterial community. Shared OTUs between groups of interest were visualised using Venn diagrams generated using the program Venny[69].

Shannon diversity was calculated for each life stage and environment samples. We used ANOVA to compare Shannon diversity of different samples and Tukey's test to identify which groups differed significantly. A Bray–Curtis distance matrix was used to calculate beta diversity of samples and visualised using a non-metric multidimensional scaling (NMDS) plot. The relative contribution of life stage and population in structuring the microbiome was assessed using permutational multivariate analysis of variance (PERMANOVA)[70] using the adonis function in the Vegan[71] package in R. PERMANOVA was also carried out to compare community structure of larvae and metamorphs with environmental samples. Analysis was performed for combined populations to isolate the effect of site as well as for individual populations to compare between site effects.

**Microbiota in epizootic and enzootic metamorph populations**. To determine the extent to which metamorph skin bacterial communities differed between populations, Shannon diversity was calculated and ANOVA was used to determine significant differences between groups. A post-hoc Tukey's test was carried out to identify populations that differed significantly.

Beta diversity of populations was calculated using a Bray–Curtis distance matrix and visualised using NMDS plots. PERMANOVA was carried out to determine the relative effects of disease dynamic and population in shaping bacterial communities. Indicator analysis[72] was performed using the labdsv package[73] in R to identify OTUs that best explained differences between populations and disease states. Indicator analysis calculates the product of an OTU's frequency and relative abundance in predefined groups. An indicator value of 1 represents the presence of an OTU in one group but not others, while an indicator value of zero is indicative of an OTU evenly distributed across all groups. Multiple comparisons were taken into account using the false discovery rate procedure[74]. OTUs with an indicator value >0.5 and q-value < 0.05 were considered informative. A heatmap displaying the relative abundance of the top 20 OTUs for enzootic and epizootic populations was generated using the function heatmap.2 in the gplots[75] package in R.

To examine whether function of bacterial communities from enzootic and epizootic metamorph populations differed, gene prediction was performed using Piphillin[76]. A sequence identity cut-off of 97% was used and sequences were assigned putative function using the KEGG reference database (May 2017 release). Beta diversity of KEGG orthology (KO) abundances was calculated using the Bray–Curtis metric. Comparison of functional profiles of each population was performed using NMDS ordination and PERMANOVA in the R packages Phyloseq[65] and Vegan[71]. Differentially abundant features between enzootic and epizootic populations were determined using linear discriminant analysis (LDA) effect size (LEfSe)[77]. Classes were defined as enzootic or epizootic with population as a subclass and an LDA score of ≥2.0 was used as a cut-off.

**Bd isolation and genome sequencing**. Bd was isolated from A. obstetricans toe clips taken from each population between 2004 and 2015. A summary of the isolates used in the study is shown in Supplementary Data 3. Isolates were grown in 50 ml Nalgene Nunc™ tissue culture-treated flasks (Thermo Fisher Scientific, Massachusetts, USA) for 10–14 days at 18–20 °C. DNA extraction was performed using the MasterPure™ Yeast DNA Purification Kit (Epicentre, Wisconsin, USA) or Qiagen Genomic Tips 20/G and DNeasy Blood and Tissue Kits (Qiagen, Venlo, Netherlands). DNA extractions were quantified using a Tapestation 2200 (Agilent Technologies, California, United States) and Qubit 2.0 fluorometer (Thermo Fisher

Scientific, Massachusetts, USA). We prepared DNA samples for sequencing on an Illumina HiSeq 2000 (Illumina, California, USA).

Raw sequencing reads were first cleaned of adapter sequences and quality trimmed using cutadapt v1.10[78]. Reads were mapped to the JEL423 reference genome (GenBank assembly accecssion: GCA_000149865.1) using Burrows−Wheeler Aligner (BWA-mem) v0.7.8[79]. Resulting sequence alignment/map (SAM) files were processed using SAMtools v1.3.1[80] using the 'fixmate' and 'sort' programs to ready the files for variant discovery. Variant discovery was done using freebayes version dbb6160[81]. In the first step, sorted BAM files for each of the isolates in the study were independently called to find variant positions. The set of all variable positions identified across all samples were merged into a single variant call format (VCF) file. In the second step, genotype calls were independently made for each isolate, at each of the positions identified in the first step to produce a squared-off call set (each sample VCF has genotype calls at the same loci, including homozygous reference calls, and explicitly identifying positions without sequencing read coverage). The sample VCF files were processed by vcflib[82] to break complex variants into allelic primitives and then to normalise short insertion and deletion sequences (indels). VCFs were then quality filtered with bcftools version 1.3.1[80]. Sites with homozygous reference genotypes covered by four reads, or sites not covered by any reads were set to missing. Subsequently, putative non-reference genotype calls were set to homozygous reference if they failed any of the following filters: there was not enough evidence to support a variant genotype call; an alternate allele is in the called genotype without supporting reads ('AC > 0 && NUMALT == 0). The phred-scaled quality score is less than 5 when there are any reads covering the position (%QUAL < 5 && DP > 0); any called allele is not supported by at least two reads (AF[*] <= 0.5 && DP < 4); alternate alleles are supported by only low-quality reads (AF[*] <= 0.5 && DP < 13 && %QUAL < 10)| (AF[*] > 0.5 && DP < 4 && %QUAL < 50); the quality, scaled by depth of supporting reads is less than a threshold (%QUAL / AO < 10); a called allele does not appear on both forward and reverse strands (SAF == 0 | SAR == 0); alleles are only supported by reads entirely placed right or left of the query variant (RPR = 0 | RPL = 0). The individual filtered VCF files were merged into a single multi-sample VCF using vcfstreamsort[82] to ensure sorting of out of order variants resulting from breaking complex variants in individual files into their simplest allelic representation. A phylogeny of the Bd isolates used in this study was visualised using ggtree[83].

Principle component analysis was carried out using bi-allelic SNP positions using the package SNPrelate v1.10.2[84] in R version 3.4.1. SNPs in high linkage disequilibrium (LD) were pruned using a threshold of approximately half the maximum value of LD in BdGPL (an LD50 of 0.125). A subset of 771 from 50,561 SNPs were used in the PCA analysis. PCA data was plotted using ggplot2[85].

**Data availability**. Sequence data have been deposited on the BioProject database under accession codes PRJNA421328 and PRJNA413876. All other data are available upon request from the authors.

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

## Acknowledgements

Dirk Schmeller and Adeline Loyau were invaluable in their support and organisation of permits. Jennifer Shelton, Nick Grassly and Maryam Grassly provided dedicated field support. We thank the community of Lescun and Parc National de Pyrenees for continued support. K.A.B. was funded by a CASE studentship from NERC; M.C.F. and T.W. J.G. were funded by the NERC award NE/E006701/1 and the Biodiversa project RACE: Risk Assessment of Chytridiomycosis to European Amphibian Biodiversity.

## Author contributions

K.A.B., J.B., F.C.C., O.D. and M.C.F. collected the data. K.A.B., K.H., F.C.C., E.J.M., S.O., L.B. and X.A.H. analysed the data. K.A.B., M.C.F., T.W.J.G. and X.A.H. wrote the paper.

## Additional information

**Competing interests:** The authors declare no competing financial interests.

