## [Peer Review File · Nature Communications]

Reviewers' comments:

Reviewer #1 (Remarks to the Author):

The manuscript entitled, "Bacterial community structure linked to enzootic versus epizootic dynamics across montane outbreaks of amphibian chytridiomycosis" is being considered for publication in Nature Communications. This paper presents results from a European amphibian system (midwife toads, *Alytes obstetricans*), and specifically addresses the skin bacterial communities on 5 different montane populations and how they are associated with fungal disease status of those populations. How the amphibian skin microbiome interacts with Bd (the fungal pathogen) has become a popular topic of research and has yielded new discoveries in the last 10 years, both from lab experimental studies as well as field studies, such as this one. The manuscript is well written and clearly presented, in most aspects. The analyses are mostly robust and I think the approaches used by the authors are sound.

I have three main criticisms of this paper:

1. The findings of this paper are not particularly novel, as all of the main findings have already been demonstrated, albeit for different amphibian species. The earliest study to demonstrate a field pattern by which amphibian populations persisting with Bd had different bacteria than similar populations that were declining with Bd was Lam et al. (2010). This 2010 paper directly cultured bacteria from the field sites and tested it against the strain of Bd found in the sites, thus it took a further step to demonstrate the anti-Bd activity of particular bacteria. Since then, Jani and Briggs (2014) demonstrated that particular bacterial groups shift in relative abundance in response to Bd infection and they used natural field populations coupled with manipulative lab experiments. The present study by Bates et al. does contribute a new field system and a different species, but the field patterns they demonstrate are not new. Additionally, several lab experiments have clearly demonstrated links between amphibian skin bacterial community composition and Bd infection (e.g., Harris et al. 2009, Becker et al. 2015, Kueneman et al. 2016).

The other findings presented here by Bates et al. regarding the life history of amphibians and the skin microbiome have also been previously reported in other systems. The finding that the skin microbiome differs pre and post metamorphosis was demonstrated in Kueneman et al. (2014, 2015). The finding that the amphibian skin microbiome differs from the environment has also been demonstrated previously: Walke et al. 2014. The present manuscript does a fine job citing much of the relevant previous work, however, my intent is to underscore that the main findings here are adding another system to the growing body of work that supports a link between the skin microbiome and Bd, but it is not new. The present paper did examine Bd strain variation across the sites, but they found that the Bd strains were the same across populations. If they had found that the Bd strains differed and that the effect of the skin microbiome was dependent upon the Bd strain type, that would have been a novel finding, for example. -But it would have also necessitated the use of functional assays of the bacteria in the system (see next section).

Becker, M.H., Walke, J.B., Cikanek, S., Savage, A.E., Mattheus, N., Santiago, C.N., Minbiole, K.P., Harris, R.N., Belden, L.K. and Gratwicke, B., 2015. Composition of symbiotic bacteria predicts survival in Panamanian golden frogs infected with a lethal fungus. *Proceedings of the Royal Society of London B: Biological Sciences*, 282(1805), p.20142881.

Harris, R.N., Brucker, R.M., Walke, J.B., Becker, M.H., Schwantes, C.R., Flaherty, D.C., Lam, B.A., Woodhams, D.C., Briggs, C.J., Vredenburg, V.T. and Minbiole, K.P., 2009. Skin microbes on frogs prevent morbidity and mortality caused by a lethal skin fungus. *The ISME journal*, 3(7), pp.818-824.

Jani, A.J. and Briggs, C.J., 2014. The pathogen *Batrachochytrium dendrobatidis* disturbs the frog skin microbiome during a natural epidemic and experimental infection. *Proceedings of the National Academy of Sciences*, 111(47), pp.E5049-E5058.

Kueneman, J.G., Woodhams, D.C., Harris, R., Archer, H.M., Knight, R. and McKenzie, V.J., 2016, September. Probiotic treatment restores protection against lethal fungal infection lost during amphibian captivity. In *Proc. R. Soc. B* (Vol. 283, No. 1839, p. 20161553). The Royal Society.

Kueneman, J.G., Woodhams, D.C., Van Treuren, W., Archer, H.M., Knight, R. and McKenzie, V.J., 2015. Inhibitory bacteria reduce fungi on early life stages of endangered Colorado boreal toads (*Anaxyrus boreas*). *The ISME journal*, 10(4), pp.934-944.

Kueneman, J.G., Parfrey, L.W., Woodhams, D.C., Archer, H.M., Knight, R. and McKenzie, V.J., 2014. The amphibian skin-associated microbiome across species, space and life history stages. *Molecular Ecology*, 23(6), pp.1238-1250.

Lam, B.A., Walke, J.B., Vredenburg, V.T. and Harris, R.N., 2010. Proportion of individuals with anti-*Batrachochytrium dendrobatidis* skin bacteria is associated with population persistence in the frog *Rana muscosa*. *Biological Conservation*, 143(2), pp.529-531.

Walke, J.B., Becker, M.H., Loftus, S.C., House, L.L., Cormier, G., Jensen, R.V. and Belden, L.K., 2014. Amphibian skin may select for rare environmental microbes. *The ISME journal*, 8(11), pp.2207-2217.

Woodhams, D.C., et al. 2014. Interacting symbionts and immunity in the amphibian skin mucosome predict disease risk and probiotic effectiveness. *PLoS One*, 9(4), p.e96375.

Woodhams, D.C., Alford, R.A., Antwis, R.E., Archer, H., Becker, M.H., Belden, L.K., Bell, S.C., Bletz, M., Daskin, J.H., Davis, L.R. and Flechas, S.V., 2015. Antifungal isolates database of amphibian skin-associated bacteria and function against emerging fungal pathogens. *Ecology*, 96(2), pp.595-595.

2. The links between the skin bacterial communities and fungal disease state of the populations in this study are strictly correlative. Microbiome studies are now going further than 16S marker gene survey data, and getting more direct measures of bacterial function in these systems, yet the findings of this study rely on only 16S marker gene bacterial community profiling. Several other studies of the amphibian skin microbiome are coupling 16S with culture-based work (e.g., bacteria versus *Bd* challenge assays in vitro, see Woodhams et al. 2015), mucosome testing (e.g., Woodhams et al. 2014), etc. Without any functional assays of the bacteria in this study, all of the discussion is very speculative. There are clearly differences in the amphibian skin bacteria between sites in the Bates et al. dataset, however, there is no basis to discern whether any functional traits of bacteria are responding to the disease state of the host population - particularly because all 5 populations have different bacteria that are driving the differences (Fig 4). For example, lines 247-257 discuss an OTU from the Lhurs site that is identified as belonging to the genus *Lysobacter* - and the authors cite other work that discusses specific strains of *Lysobacter* that are known to inhibit *Bd*. However, it is widely known that the short sequence reads obtained from 16S are not reliable to distinguish species or strain level differences, nor to then infer functional aspects of those bacteria. It is plausible that the *Lysobacteria* found in Lhurs has no inhibitory activity against *Bd*, but it can't be known from these data. Same comment applies to lines 279-280, where the authors discuss whole Orders of bacteria and make assumptions about their source and function - which is not appropriate. Unfortunately, that is the limitation of the scope of approaches used in this paper. The authors acknowledge this limitation at the end of the discussion section: 333-339.

3. The status of the 5 populations is characterized as epizootic or enzootic, but the data provided to support those delineations is not particularly strong. Without reading all of the previous work of these authors on these particular sites (which are part of a long term monitoring system), it is not very clear how the sites are characterized as epizootic and enzootic. In Figure 2, the sample sizes that were tested are not provided per site. The caption states "n = 32 (Enzootic); n = 19 (Epizootic)". I interpret this to mean that 19 total amphibians were swabbed from the two epizootic sites (Ansabere and Arlet) combined, meaning that probably 10 or less than 10 individuals were tested from each site. These are very low numbers to conduct cross-site comparisons of Bd infection intensity and prevalence, as Bd can be highly variable and is likely to be aggregated. Sample sizes per site should be 20 at a minimum and better if there are 30 or more. I do not see where the authors provide statistics to support the differences among these five populations - and some of the comparisons do not look significantly different (e.g., Fig 2a: Ansabere may be the same as the enzootic populations). The first section of the supplemental information discusses the disease status of these populations but the critical citation that is provided is from an unpublished thesis - which is not a robust source. Perhaps those data are in preparation for a separate publication, but it leaves the current manuscript weak on this issue. Lastly, the bacterial profiles indicate that these five sites are unique from each other (Figure 4 and 3b), regardless of the purported disease state. Given that there are only 5 sites, it is hard to argue that disease state is the critical factor driving these site differences.

Reviewer #2 (Remarks to the Author):

This is a very clear and well-written manuscript that relates cutaneous bacterial community structure of a frog species to population, life-history stage and disease status of the population (epizootic or enzootic). The results are clear, and the authors' interpretations are on target.

The biology of bacterial-fungal interactions assumes great practical importance in light of fungal pathogens Bd and Bsal, both of which pose serious threats to amphibian biodiversity.

As the authors acknowledge cause and effect are not determined in this study, i.e., did variation in disease state cause variation in bacterial community structure or did variation in community structure lead to variation in disease state? However, the patterns are so clear that this survey merits publication in a high impact journal in order to stimulate additional research.

The authors make the point (317) that attention should focus on entire bacterial communities rather than on the effects of single OTUs. I agree in general with this statement, however at this point we don't know enough to know the circumstances under which single key species are under selection or when entire "community structures" are under selection. There are a number of possibilities, e.g., selection for a single OTU, which could determine community structure if the OTU is a keystone species.

I was surprised that the alpha diversity on the frogs' skins was higher than that of sediment and water.

The authors might comment on the fungal component of the skin microbiota, including fungi that might be anti-Bd.

Minor comments:

51 - there are more recent publications on immune system - bacterial interactions

72 – please define “key processes”

Reviewer #3 (Remarks to the Author):

This is a very well-written and organized manuscript that describes bacterial communities from the skin of amphibians. The authors found differences in bacterial communities structure between disease status (epizootic vs enzootic), and age classes.

I only have a couple of questions:

Line 33 bacterial communities differed significantly amongst larvae and metamorphs between disease states or just as age classes?

Line 37 genomes of Bd isolates comes out of nowhere, maybe introduce it earlier?

Lines 87-95 I’m not sure if this paragraph is appropriately placed here, since it is all about the results.

Line 115 What kind of alpha diversity measure was used? Richness? Chao index? Phylo diversity?

I would also recommend changing the verb 'linked' in the title since this paper analyses samples from wild populations and it is not an experiment. maybe a word like 'correlates'?

Reviewers' comments:

Reviewer #1 (Remarks to the Author):

The manuscript entitled, "Bacterial community structure linked to enzootic versus epizootic dynamics across montane outbreaks of amphibian chytridiomycosis" is being considered for publication in Nature Communications. This paper presents results from a European amphibian system (midwife toads, *Alytes obstetricans*), and specifically addresses the skin bacterial communities on 5 different montane populations and how they are associated with fungal disease status of those populations. How the amphibian skin microbiome interacts with Bd (the fungal pathogen) has become a popular topic of research and has yielded new discoveries in the last 10 years, both from lab experimental studies as well as field studies, such as this one. The manuscript is well written and clearly presented, in most aspects. The analyses are mostly robust and I think the approaches used by the authors are sound.

I have three main criticisms of this paper:

- 1.** The findings of this paper are not particularly novel, as all of the main findings have already been demonstrated, albeit for different amphibian species. The earliest study to demonstrate a field pattern by which amphibian populations persisting with Bd had different bacteria than similar populations that were declining with Bd was Lam et al. (2010). This 2010 paper directly cultured bacteria from the field sites and tested it against the strain of Bd found in the sites, thus it took a further step to demonstrate the anti-Bd activity of particular bacteria. Since then, Jani and Briggs (2014) demonstrated that particular bacterial groups shift in relative abundance in response to Bd infection and they used natural field populations coupled with manipulative lab experiments. The present study by Bates et al. does contribute a new field system and a different species, but the field patterns they demonstrate are not new. Additionally, several lab experiments have clearly demonstrated links between amphibian skin bacterial community composition and Bd infection (e.g., Harris et al. 2009, Becker et al. 2015, Kueneman et al. 2016).

The other findings presented here by Bates et al. regarding the life history of amphibians and the skin microbiome have also been previously reported in other systems. The finding that the skin microbiome differs pre and post metamorphosis was demonstrated in Kueneman et al. (2014, 2015). The finding that the amphibian skin microbiome differs from the environment has also been demonstrated previously: Walke et al. 2014. The present manuscript does a fine job citing much of the relevant previous work, however, my intent is to underscore that the main findings here are adding another system to the growing body of work that supports a link between the skin microbiome and Bd, but it is not new. The present paper did examine Bd strain variation across the sites, but they found that the Bd strains were the same across

populations. If they had found that the *Bd* strains differed and that the effect of the skin microbiome was dependent upon the *Bd* strain type, that would have been a novel finding, for example. -But it would have also necessitated the use of functional assays of the bacteria in the system (see next section).

We recognise the importance of prior studies in enhancing our current understanding of the microbial ecology of amphibian skin, however the literature cited by Reviewer 1 are not without limitations that are better dealt with in our manuscript. Lam *et al.* (2010) studied only two populations and therefore were limited in their ability to disentangle population versus disease effects. In addition they profiled bacteria using only culture based methods and therefore captured only a fraction of the bacterial taxonomic diversity that were culturable. While Lam *et al.* (2010) did in-vitro *Bd*-bacteria challenge assays on the small number of isolates cultured, these assays are limited in assigning function since bacteria are grown with *Bd* in monoculture so community effects are not taken into account. In addition, the nutrient media differs significantly from host skin and therefore bacteria are likely to respond differently in the absence of host effects.

Jani & Briggs (2014) and more recently Jani *et al.* (2017) did show associations between microbiome and disease dynamics in the lab and field, however this was performed on a single North American study system. Our work therefore differs significantly in both host species and environmental variables. Though some of the results of the current manuscript have been addressed before, we believe replication of those results in a completely different study system on a different continent represents an important contribution to the literature.

Furthermore, the principal novelty of our work is in our integration of pathogen genomics, host microbiome structure and long-term disease dynamics in a single framework. We emphasise that though prior studies have greatly enhanced our knowledge of microbial ecology of amphibian skin, no such study has been able to simultaneously account for pathogen genotypic variation on disease outcome. Importantly, in the Sierra Nevada system used by Jani & Briggs (2014) and Jani *et al.* (2017), a prior study has shown a strong association between the pathogen (*Bd* isolate) and disease dynamics (enzootic or epizootic) (Piova-Scott *et al.* 2015) but no study to date has investigated the relative importance of the microbiome and pathogen genetics simultaneously. Our study is the first of its kind to take into account both pathogen genetics and host microbiome. Further, by demonstrating that pathogen genetics are not associated with disease outcome, while a strong link between disease and the microbiome exists, we consider our findings highly novel.

Becker, M.H., Walke, J.B., Cikanek, S., Savage, A.E., Mattheus, N., Santiago, C.N., Minbiole, K.P., Harris, R.N., Belden, L.K. and Gratwicke, B., 2015. Composition of symbiotic bacteria predicts survival in Panamanian golden frogs infected with a lethal fungus. *Proceedings*

of the Royal Society of London B: Biological Sciences, 282(1805), p.20142881.

Harris, R.N., Brucker, R.M., Walke, J.B., Becker, M.H., Schwantes, C.R., Flaherty, D.C., Lam, B.A., Woodhams, D.C., Briggs, C.J., Vredenburg, V.T. and Minbiole, K.P., 2009. Skin microbes on frogs prevent morbidity and mortality caused by a lethal skin fungus. *The ISME journal*, 3(7), pp.818-824.

Jani, A.J. and Briggs, C.J., 2014. The pathogen *Batrachochytrium dendrobatidis* disturbs the frog skin microbiome during a natural epidemic and experimental infection. *Proceedings of the National Academy of Sciences*, 111(47), pp.E5049-E5058.

Jani, A.J., Knapp, R.A. and Briggs, C.J. 2017. Epidemic and endemic pathogen dynamics correspond to distinct host population microbiomes at a landscape scale. *Proc. R. Soc. B.* 284, 20170944.

Kueneman, J.G., Woodhams, D.C., Harris, R., Archer, H.M., Knight, R. and McKenzie, V.J., 2016, September. Probiotic treatment restores protection against lethal fungal infection lost during amphibian captivity. In *Proc. R. Soc. B* (Vol. 283, No. 1839, p. 20161553). The Royal Society.

Kueneman, J.G., Woodhams, D.C., Van Treuren, W., Archer, H.M., Knight, R. and McKenzie, V.J., 2015. Inhibitory bacteria reduce fungi on early life stages of endangered Colorado boreal toads (*Anaxyrus boreas*). *The ISME journal*, 10(4), pp.934-944.

Kueneman, J.G., Parfrey, L.W., Woodhams, D.C., Archer, H.M., Knight, R. and McKenzie, V.J., 2014. The amphibian skin-associated microbiome across species, space and life history stages. *Molecular Ecology*, 23(6), pp.1238-1250.

Piovia-Scott, J. et al. 2015. Correlates of virulence in a frog-killing fungal pathogen: evidence from a California amphibian decline. *The ISME Journal*, 9, 1570–1578.

Lam, B.A., Walke, J.B., Vredenburg, V.T. and Harris, R.N., 2010. Proportion of individuals with anti-*Batrachochytrium dendrobatidis* skin bacteria is associated with population persistence in the frog *Rana muscosa*. *Biological Conservation*, 143(2), pp.529-531.

Walke, J.B., Becker, M.H., Loftus, S.C., House, L.L., Cormier, G., Jensen, R.V. and Belden, L.K., 2014. Amphibian skin may select for rare environmental microbes. *The ISME journal*, 8(11), pp.2207-2217.

Woodhams, D.C., et al. 2014. Interacting symbionts and immunity in the amphibian skin mucosome predict disease risk and probiotic effectiveness. *PLoS One*, 9(4), p.e96375.

Woodhams, D.C., Alford, R.A., Antwis, R.E., Archer, H., Becker, M.H., Belden, L.K., Bell, S.C.,

Bletz, M., Daskin, J.H., Davis, L.R. and Flechas, S.V., 2015. Antifungal isolates database of amphibian skin-associated bacteria and function against emerging fungal pathogens. *Ecology*, 96(2), pp.595-595.

2. The links between the skin bacterial communities and fungal disease state of the populations in this study are strictly correlative. Microbiome studies are now going further than 16S marker gene survey data, and getting more direct measures of bacterial function in these systems, yet the findings of this study rely on only 16S marker gene bacterial community profiling. Several other studies of the amphibian skin microbiome are coupling 16S with culture-based work (e.g., bacteria versus *Bd* challenge assays in vitro, see Woodhams et al. 2015), mucosome testing (e.g., Woodhams et al. 2014), etc. Without any functional assays of the bacteria in this study, all of the discussion is very speculative. There are clearly differences in the amphibian skin bacteria between sites in the Bates et al. dataset, however, there is no basis to discern whether any functional traits of bacteria are responding to the disease state of the host population - particularly because all 5 populations have different bacteria that are driving the differences (Fig 4). For example, lines 247-257 discuss an OTU from the Lhurs site that is identified as belonging to the genus *Lysobacter* - and the authors cite other work that discusses specific strains of *Lysobacter* that are known to inhibit *Bd*. However, it is widely known that the short sequence reads obtained from 16S are not reliable to distinguish species or strain level differences, nor to then infer functional aspects of those bacteria. It is plausible that the *Lysobacteria* found in Lhurs has no inhibitory activity against *Bd*, but it can't be known from these data. Same comment applies to lines 279-280, where the authors discuss whole Orders of bacteria and make assumptions about their source and function - which is not appropriate. Unfortunately, that is the limitation of the scope of approaches used in this paper. The authors acknowledge this limitation at the end of the discussion section: 333-339.

We agree with Reviewer 1 that direct functional information would be beneficial for determining whether differences in taxonomic composition relate to functional differences in susceptibility. Indeed, characterisation of bacterial function via metabolomics is already underway for this study system and will form the basis of a future investigation. Consequently, direct functional characterisation was not in the scope of this study which was primarily investigating the interplay between pathogen genetics, microbial community structure and disease outcome. The methods used to investigate bacterial function in previous papers have been informative in our understanding of the interactions between single bacterial isolates and *Bd*, however we feel that methods that take into account bacterial function in complex communities would be more appropriate for our study. In addition, while the mucosome assay developed by Woodhams *et al.* (2014) provides a holistic measure of amphibian skin defences, it doesn't differentiate between bacterial derived metabolites and anti-microbial defences produced by the host. Both figure 4 and SI table 5 demonstrate that

there is a subset of taxa that are common to populations exhibiting a specific disease dynamic. To gain some functional insight into the bacterial communities of *A. obstetricans* metamorphs from enzootic and epizootic populations we have assigned putative functional traits to OTUs based on OTU sequence identities using the program Piphillin (Iwai *et al.* 2016). A program based on a similar algorithm (PiCrust) has been used in recent papers for a range of systems including amphibians (Bletz *et al.* 2016, Ziegler *et al.* 2017). While we agree with the reviewer that this method is limited in not measuring direct functional information in the same way that metabolomics or metatranscriptomics does, it does demonstrate a significant difference in predicted function based on population and disease dynamic (SI Fig. 4) and therefore lends support to the overall findings of this study.

We agree that there is no way of determining whether specific taxa such as *Lysobacter* are *Bd* inhibitory using the methods in this study. However, we feel that highlighting taxa that show strong associations with a particular disease state that have shown similar patterns in prior studies is beneficial to the reader in laying a framework for future hypotheses. This is also common practice in many publications e.g. Rebollar *et al.* (2016).

Bletz, M.C., Goedbloed, D.J., Sanchez, E., Reinhardt, T., Tebbe, C.C., Bhujju, S., Geffers, R., Jarek, M., Vences, M., Steinfartz, S. 2016. Amphibian gut microbiota shifts differentially in community structure but converges on habitat specific predicted functions. *Nature Communications*, 7, 13699.

Iwai, S. *et al.* 2016. Piphillin: Improved Prediction of Metagenomic Content by Direct Inference from Human Microbiomes. *PLoS ONE*, 11, e0166104.

Rebollar, E.A. *et al.* 2016. Skin bacterial diversity of Panamanian frogs is associated with host susceptibility and presence of *Batrachochytrium dendrobatidis*. *ISME Journal*, 10, 1682-1695.

Ziegler, M., Seneca, F.O., Yum, L.K., Palumbi, S.R., Voolstra, C.R. 2017. Bacterial community dynamics are linked to patterns of coral heat tolerance. *Nature Communications*, 8, 14213.

3. The status of the 5 populations is characterized as epizootic or enzootic, but the data provided to support those delineations is not particularly strong. Without reading all of the previous work of these authors on these particular sites (which are part of a long term monitoring system), it is not very clear how the sites are characterized as epizootic and enzootic. In Figure 2, the sample sizes that were tested are not provided per site. The caption states "n = 32 (Enzootic); n = 19 (Epizootic)". I interpret this to mean that 19 total amphibians were swabbed from the two epizootic sites (Ansabere and Arlet) combined, meaning that probably 10 or less than 10 individuals were tested from each site. These are very low numbers to conduct cross-site comparisons of *Bd* infection intensity and prevalence, as *Bd* can be highly variable and is likely to be aggregated.

Sample sizes per site should be 20 at a minimum and better if there are 30 or more. I do not see where the authors provide statistics to support the differences among these five populations - and some of the comparisons do not look significantly different (e.g., Fig 2a: Ansabere may be the same as the enzootic populations). The first section of the supplemental information discusses the disease status of these populations but the critical citation that is provided is from an unpublished thesis - which is not a robust source. Perhaps those data are in preparation for a separate publication, but it leaves the current manuscript weak on this issue. Lastly, the bacterial profiles indicate that these five sites are unique from each other (Figure 4 and 3b), regardless of the purported disease state. Given that there are only 5 sites, it is hard to argue that disease state is the critical factor driving these site differences.

We thank Reviewer 1 for pointing out this opportunity to strengthen this portion of the manuscript. We have extended our dataset of larval population abundance and metamorph infection data from three to six years and more clearly explain the distinction between enzootic and epizootic sites on lines 139-147. Specifically we define epizootic sites as showing declining population abundance in the presence of *Bd* and enzootic sites as showing stable population abundance whilst maintaining sustained *Bd* infection. Line 153-154 and Figure 2 also show a significant difference in infection intensity based on disease dynamic.

We have updated figure 2 to include sample sizes for each population. While a larger sample size for Ansabere would have been ideal, the significant population crash due to *Bd* meant we were unable to collect additional metamorphs. Despite the small sample size for Ansabere in 2015, we feel our distinction of sites as enzootic and epizootic is justified given the long-term disease data.

We have provided statistics supporting the significant differences in microbiome based on population and disease dynamic on lines 158-161. Figure 3B supports disease state as driving differences in microbiome by showing closer clustering of Ansabere and Arlet which both share a closer geographic proximity to enzootic populations yet exhibit similar disease dynamics. In addition, figure 4 and SI Table 5 show several OTUs that are common to populations based on disease dynamic, in particular *Novosphingobium* and *Planococcaceae* in the enzootic populations and *Chryseobacterium* in the Epizootic populations.

Reviewer #2 (Remarks to the Author):

This is a very clear and well-written manuscript that relates cutaneous bacterial community structure of a frog species to population, life-history stage and disease status of the population (epizootic or enzootic). The results are clear, and the authors' interpretations are on target.

The biology of bacterial-fungal interactions assumes great practical importance in light of fungal pathogens *Bd* and *Bsal*, both of which pose serious threats to amphibian biodiversity.

As the authors acknowledge cause and effect are not determined in this study, i.e., did variation in disease state cause variation in bacterial community structure or did variation in community structure lead to variation in disease state? However, the patterns are so clear that this survey merits publication in a high impact journal in order to stimulate additional research.

The authors make the point (317) that attention should focus on entire bacterial communities rather than on the effects of single OTUs. I agree in general with this statement, however at this point we don't know enough to know the circumstances under which single key species are under selection or when entire "community structures" are under selection. There are a number of possibilities, e.g., selection for a single OTU, which could determine community structure if the OTU is a keystone species.

We agree with Reviewer 2 that a single species may exert strong effects on the overall community. However, given the findings from this study in which differences in beta diversity and bacterial community composition based on disease dynamic were found, we feel that future studies investigating multispecies interactions (which may in turn identify keystone species) are likely to be most informative.

I was surprised that the alpha diversity on the frogs' skins was higher than that of sediment and water.

The authors might comment on the fungal component of the skin microbiota, including fungi that might be anti-*Bd*.

We fully agree that there is wider need for researchers in this field to consider microbial communities other than bacteria when studying host-microbe interactions in the context of disease. We have included a comment in our discussion highlighting the need to expand the kingdoms studied in microbiome studies with particular reference to Fungi (lines 368-369). Research on the link between bacterial and fungal communities and *Bd* dynamics is currently being undertaken and will form the basis of a future paper.

Minor comments:

51 – there are more recent publications on immune system – bacterial interactions

We have included a more recent paper by Gensollen *et al.* (2016) (line 51).

Gensollen, T., Iyer, S.S., Kasper, D.L. & Blumberg, R.S. 2016. How colonization by microbiota in early life shapes the immune system. *Science* **352**, 539-544.

72 – please define “key processes”

Clarified. We have reworded this to "examine" as this implies a more pattern based approach which is in line with the scope of this manuscript.

“Consequently such studies are limited in their ability to examine the relationships between host bacterial communities and disease dynamics in complex ecosystems”

Reviewer #3 (Remarks to the Author):

This is a very well-written and organized manuscript that describes bacterial communities from the skin of amphibians. The authors found differences in bacterial communities structure between disease status (epizootic vs enzootic), and age classes.

I only have a couple of questions:

Line 33 bacterial communities differed significantly amongst larvae and metamorphs between disease states or just as age classes?

Our life stage analysis compared age classes only, not disease state as pathology occurs at metamorphosis and we did not have larval data for one epizootic population.

Line 37 genomes of *Bd* isolates comes out of nowhere, maybe introduce it earlier?

Thank you for identifying this opportunity to improve the clarity of the introduction. We have updated our abstract and introduction to better introduce the *Bd* genomics.

Lines 87-95 I'm not sure if this paragraph is appropriately placed here, since it is all about the results.

We have decided to keep this paragraph as we feel it gently introduces the complex findings of the paper that are explained in greater detail in the Results and Discussion sections.

Line 115 What kind of alpha diversity measure was used? Richness? Chao index? Phylo diversity?

We used Shannon Diversity, Methods, Line 463

I would also recommend changing the verb 'linked' in the title since this paper analyses samples from wild populations and it is not an experiment. maybe a word like 'correlates'?

We fully agree with Reviewer 3 that it would be wrong to imply causation here, however we have decided to keep “linked” as it keeps the title more general. We have also changed the title to highlight the pathogen genomics component.

Reviewers' comments:

Reviewer #1 (Remarks to the Author):

The revised version of this manuscript is improved in several ways - especially in terms of providing more substantial background for how the sites were designated as either epizootic and enzootic. The authors also added several more citations and descriptions to demonstrate how their work builds on previous systems. I agree with the authors' response that demonstrating this pattern in a different amphibian system (on a different continent) is of great value in further supporting a strong role of the microbiome in the amphibian skin pathogen system.

The one area that I do think can use further revision has to do with how the authors are spinning the use of the pathogen genomics. Yes, it is true that it is a novel approach to combine pathogen genomics and microbiome study in a field system - and this is better emphasized in the revised version. However, I think the authors go too far in claiming that the microbiome and NOT pathogen genotype is the main driver - for the simple reason that there was only one pathogen genotype in this system. In order to claim that the microbiome is more important than pathogen genotype, the system would need to involve multiple pathogen genotypes to tease that apart. Thus, I think the new title is misleading on this issue. A more appropriate title would be, "Bacterial community structure is linked to enzootic versus epizootic dynamics across montane outbreaks caused by a single genotype of the amphibian chytrid fungus" - terrible, I know, but just to illustrate the difference.

The same logic applies to the revised sentence on lines 216-218, which reads, "In addition for the first time we investigate the relationship between Bd genotype and the microbiome in disease outcome by sequencing Bd isolates from each population." This is misleading. Because there was only one Bd genotype, the authors are not able to investigate this relationship. I suggest removing this type of language and replacing it with a brief discussion of what the authors might expect if a system DID involve more than one Bd genotype. That would be far more appropriate.

Reviewer #2 (Remarks to the Author):

The authors have done an excellent job of addressing reviewers' comments. While not experimental, the study is very strong in identifying links between life history, disease status and bacterial community structure. Links to bacterial community function are more inferential and are considered preliminary. However, the authors could add a sentence or two commenting on the predicted functional differences and what they could mean (line 196).

The results are sufficiently novel. Having the pathogen genotyped across populations and assessing bacterial community structure is novel and was necessary for the authors to conclude that pathogen genetics were in fact not associated with disease outcome in their system. This left the association of bacterial community structure and disease to be potentially causative, although an experiment would be necessary to sort out cause and effect.

Reviewer #3 (Remarks to the Author):

I reviewed this manuscript before (R # 3) and I think the authors did a good job addressing the comments of the reviewers. Now the genomic analyses seem better incorporated since the beginning including title, intro, etc.

I recommend this paper to be published.

Reviewer #4 (Remarks to the Author):

Manuscript 127699_1 "Bacterial community structure but not pathogen genotype is linked to enzootic versus epizootic dynamics across montane outbreaks of amphibian chytridiomycosis" by Bates et al.

Review of technical aspects of whole genome sequencing of the fungal pathogen (*Batrachochytrium dendrobatidis*, Bd)

The authors performed whole genome sequencing of a number of Bd samples, and mapped these sequence reads as well as several that seem to be obtained from other studies (OTHER in Table S8? This is not quite clear.) to a Bd reference genome, and did variant calling to test for differences between samples/populations. The methods used appear appropriate; however, it is not quite possible to judge the validity of the results, because some information is missing. This concerns the following information, which the authors can easily add to supplementary table S8 or the description of Bd sequencing in the supplementary methods:

1. How many reads (of what length) were obtained for each sample?
2. How many of the (trimmed) reads mapped to the reference genome?
3. How many variants (raw/filtered) were obtained for each sample?
4. The SRA accession numbers for each sample should be given. The text at line 510 just states that the data were submitted to the SRA, but does not give an accession number. It would be good to give, for example, the BioProject accession in the text and individual accession numbers for the samples in Table S8. If data were used from previous studies (OTHER in Table S8?), then accession numbers and citations should also be given in Table S8.
5. Please give a citation and/or accession number for the Bd reference genome JEL423.

Reviewer #1 (Remarks to the Author):

The revised version of this manuscript is improved in several ways - especially in terms of providing more substantial background for how the sites were designated as either epizootic and enzootic. The authors also added several more citations and descriptions to demonstrate how their work builds on previous systems. I agree with the authors' response that demonstrating this pattern in a different amphibian system (on a different continent) is of great value in further supporting a strong role of the microbiome in the amphibian skin pathogen system.

The one area that I do think can use further revision has to do with how the authors are spinning the use of the pathogen genomics. Yes, it is true that it is a novel approach to combine pathogen genomics and microbiome study in a field system - and this is better emphasized in the revised version. However, I think the authors go too far in claiming that the microbiome and NOT pathogen genotype is the main driver - for the simple reason that there was only one pathogen genotype in this system. In order to claim that the microbiome is more important than pathogen genotype, the system would need to involve multiple pathogen genotypes to tease that apart. Thus, I think the new title is misleading on this issue. A more appropriate title would be, "Bacterial community structure is linked to enzootic versus epizootic dynamics across montane outbreaks caused by a single genotype of the amphibian chytrid fungus" - terrible, I know, but just to illustrate the difference.

We agree with Reviewer 1 that comparing *Bd* genotypes with known differences in virulence would be beneficial for disentangling pathogen-microbiome interactions. However, we feel that this may be better tested in an experimental system and thus was not within the scope of our study which investigates pathogen-microbiome dynamics over discrete spatial scales in the wild. Our genomic analyses demonstrate that *Bd* in the Pyrenees is the result of a single introduction that has since radiated across multiple lake systems. Given the single nucleotide polymorphism (SNP) differences between isolates (Fig. 5) we cannot conclude that there is a single *Bd* genotype across all lakes, but rather highly similar genetic variants that show no evidence of spatial structuring or associations with epidemiological trends. In light of these comments we have rephrased the pathogen genomics component of the paper to "pathogen genetics" rather than explicitly mentioning genotype. Given the comments made by the editor and Reviewer 1 we have also changed the title of the manuscript to "Amphibian chytridiomycosis outbreak dynamics are linked with host skin bacterial community structure".

The same logic applies to the revised sentence on lines 216-218, which reads, "In addition for the first time we investigate the relationship between *Bd* genotype and the microbiome in disease outcome by sequencing *Bd* isolates from each population." This is misleading. Because there was only one *Bd* genotype, the authors are not able to investigate this relationship. I suggest removing this type of language and replacing it with a brief discussion of what the authors might expect if a system DID involve more than one *Bd* genotype. That would be far more appropriate.

We thank Reviewer 1 for highlighting this point. We have made this clearer by not referring explicitly to *Bd* genotypes but instead to pathogen genetics. We have changed line 218 to "In addition, for the first time we investigate the relationship between *Bd*

genetics and the microbiome in disease outcome by sequencing *Bd* isolates from each population." We discuss the potential effect of multiple *Bd* genotypes on line 334.

Reviewer #2 (Remarks to the Author):

The authors have done an excellent job of addressing reviewers' comments. While not experimental, the study is very strong in identifying links between life history, disease status and bacterial community structure. Links to bacterial community function are more inferential and are considered preliminary. However, the authors could add a sentence or two commenting on the predicted functional differences and what they could mean (line 196).

The results are sufficiently novel. Having the pathogen genotyped across populations and assessing bacterial community structure is novel and was necessary for the authors to conclude that pathogen genetics were in fact not associated with disease outcome in their system. This left the association of bacterial community structure and disease to be potentially causative, although an experiment would be necessary to sort out cause and effect.

We thank Reviewer 2 for their positive comments. We have expanded our discussion on the results of predicted function on line 258: "Importantly, differences in bacterial community structure also reflect variation in bacterial predicted function. This finding is not indicative of functional redundancy among disparate bacterial communities and supports an active role of bacteria in host defence".

Reviewer #3 (Remarks to the Author):

I reviewed this manuscript before (R # 3) and I think the authors did a good job addressing the comments of the reviewers. Now the genomic analyses seem better incorporated since the beginning including title, intro, etc.

I recommend this paper to be published.

Reviewer #4 (Remarks to the Author):

Manuscript 127699_1 "Bacterial community structure but not pathogen genotype is linked to enzootic versus epizootic dynamics across montane outbreaks of amphibian chytridiomycosis" by Bates et al.

Review of technical aspects of whole genome sequencing of the fungal pathogen (*Batrachochytrium dendrobatidis*, *Bd*)

The authors performed whole genome sequencing of a number of *Bd* samples, and mapped these sequence reads as well as several that seem to be obtained from other studies (OTHER in Table S8? This is not quite clear.) to a *Bd* reference genome, and did variant calling to test for differences between samples/populations. The methods used appear appropriate; however, it is not quite possible to judge the validity of the results, because some information is missing. This concerns the following information, which

the authors can easily add to supplementary table S8 or the description of Bd sequencing in the supplementary methods:

We thank Reviewer 4 for their constructive comments. The non-Pyreanean isolates are included in Supplementary Table 8 with corresponding information regarding isolate ID, country and specific location from which they were collected included. We have included point-by-point answers to each question below.

1. How many reads (of what length) were obtained for each sample?

We have included the number of reads and their average length for each sample in SI Table 8 under the column headings “Total reads” and “Mean read length”.

2. How many of the (trimmed) reads mapped to the reference genome?

We have included this information in SI Table 8 under the column heading “Reads mapped to JEL423”.

3. How many variants (raw/filtered) were obtained for each sample?

The number of pre- and post-filtering variants (SNPs, INDELS and transitions/transversion) are included in SI Table 8. In addition we have included further sequencing information including the sequencing depth and coverage.

4. The SRA accession numbers for each sample should be given. The text at line 510 just states that the data were submitted to the SRA, but does not give an accession number. It would be good to give, for example, the BioProject accession in the text and individual accession numbers for the samples in Table S8. If data were used from previous studies (OTHER in Table S8?), then accession numbers and citations should also be given in Table S8.

The SRA numbers will be included on acceptance of publication.

5. Please give a citation and/or accession number for the Bd reference genome JEL423.

We have included the accession number for JEL423 (GenBank assembly accession: GCA_000149865.1) in the Supplementary Methods.

REVIEWERS' COMMENTS:

Reviewer #4 (Remarks to the Author):

My questions about the sequencing of Bd samples have been addressed.